# What Do
# Self-Supervised Vision Transformers Learn?

**Namuk Park**[1]* **Wonjae Kim**[2] **Byeongho Heo**[2] **Taekyung Kim**[2] **Sangdoo Yun**[2]
[1]Prescient Design, Genentech  [2]NAVER AI Lab
park.namuk@gene.com  {wonjae.kim,bh.heo,taekyung.k,sangdoo.yun}@navercorp.com

## Abstract

We present a comparative study on how and why contrastive learning (CL) and masked image modeling (MIM) differ in their representations and in their performance of downstream tasks. In particular, we demonstrate that self-supervised Vision Transformers (ViTs) have the following properties: (1) CL trains self-attentions to capture longer-range global patterns than MIM, such as the shape of an object, especially in the later layers of the ViT architecture. This CL property helps ViTs linearly separate images in their representation spaces. However, it also makes the self-attentions collapse into homogeneity for all query tokens and heads. Such homogeneity of self-attention reduces the diversity of representations, worsening scalability and dense prediction performance. (2) CL utilizes the low-frequency signals of the representations, but MIM utilizes high-frequencies. Since low- and high-frequency information respectively represent shapes and textures, CL is more shape-oriented and MIM more texture-oriented. (3) CL plays a crucial role in the later layers, while MIM mainly focuses on the early layers. Upon these analyses, we find that CL and MIM can complement each other and observe that even the simplest harmonization can help leverage the advantages of both methods.

## 1 Introduction

Contrastive Learning (CL) (He et al., 2020; Chen et al., 2020a;b; 2021) has been the most popular self-supervised learning methods until recently. It aims to learn the invariant semantics of two random views (Tian et al., 2020a;b) by making global projections of representations similar for positive samples and dissimilar for negative samples. Since CL exploits the globally projected representations to contrast each other, it can be deemed as an *"image-level"* self-supervised learning approach.

Deviating from CL, masked image modeling (MIM) (Bao et al., 2022; Xie et al., 2022b; He et al., 2022) has risen as a strong competitor of CL in the era of Vision Transformers (ViTs) (Dosovitskiy et al., 2021) with its impressive performances of downstream tasks. MIM trains ViTs by reconstructing the correct semantics of masked input patches. Unlike CL, it learns the semantics of patch tokens and this can be deemed as a *"token-level"* self-supervised learning approach. Since MIM outperforms CL in fine-tuning accuracy, it may appear *prima facie* as a more effective pre-training method than CL. However, a different trend is observed for linear probing accuracy with CL outperforming MIM (See Figure 1). For further exposition on CL and MIM, we refer the reader to Appendix B.

Then, which method—CL or MIM—should we use for the self-supervised learning of ViTs? Although both methods are widely used, little is known about what they learn. This paper sheds light on their nature by showing that ViTs trained through CL and MIM learn opposite knowledge. In particular, we raise questions to better understand self-supervised learning, and then find the answers that can potentially affect future improvements. The questions posed can be divided into the following properties of Vision Transformers: the behavior of self-attentions, the transformation of the representations, and the position of lead role components. Our key questions and findings are elaborated below.

**How do self-attentions behave? (Section 2)** We find that CL primarily captures global relationships, while MIM captures local relationships. This implies that the representations of CL contain more global patterns, such as object shapes, than those of MIM. On the one hand, this property helps

---

*Most of this work was done while the author was at NAVER AI Lab.

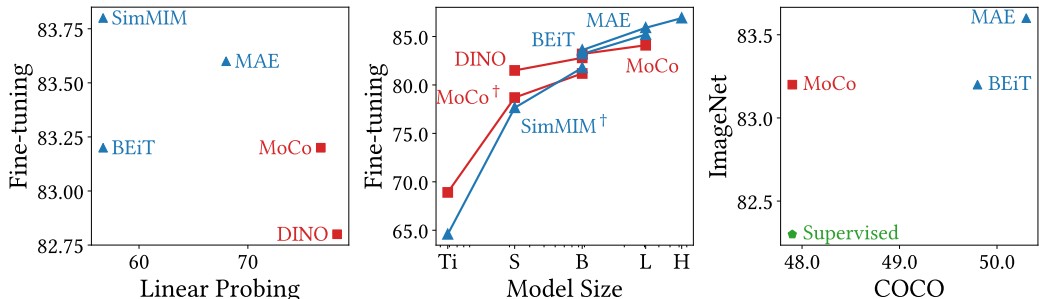

Figure 1: **CL outperforms MIM in linear probing and small model regimes.** In contrast, MIM excels in fine-tuning, large model regimes, and dense prediction. Red squares (■) denote CL, and blue triangles (▲) denote MIM. By default, we report the performance of ViT-B trained or pre-trained on ImageNet-1K. We use the results from original papers and He et al. (2022) for object detection. Regarding the scaling experiment, we report the results that we reproduced based on official configurations except with 100 epochs, marking them as MoCo[†] and SimMIM[†]. *Left*: CL outperforms MIM in linear probing but underperforms in fine-tuning. *Middle*: CL outperforms MIM in small model regimes (ViT-Ti and ViT-S), and MIM shows superior scalability in large model regimes (ViT-L and ViT-H). *Right*: MIM outperforms CL in the dense prediction downstream tasks, such as object detection with Mask R-CNN (He et al., 2017) on COCO (Lin et al., 2014).

CL recognize objects and distinguish images. On the other hand, however, it also suggests that CL struggles to preserve local information. In particular, we observe that self-attentions of CL in the later layers for all query tokens and heads collapse into homogeneous attention maps. In such cases, most self-attention maps focus on object boundaries, meaning that they can capture object shapes but may lose interaction diversity between tokens. Consequently, CL and MIM each have advantages over different tasks: CL works well for linear probing and classification tasks with smaller models, whereas MIM outperforms CL in fine-tuning and dense prediction tasks with larger models.

**How are representations transformed? (Section 3)** CL transforms representations mainly based on image-level information, and its self-attentions collect information on object shape over entire tokens. This process makes tokens similar rather than diversifying them. As a result, CL distinguishes images well but has difficulty distinguishing tokens. On the contrary, MIM preserves and amplifies token-level information. Thus, the self-attentions for each token are substantially different and prohibit each token from including redundant information. We observe the consistent property from our Fourier analysis: CL primarily utilizes the low-frequency signals, but MIM utilizes high-frequencies. This observation suggests that CL is shape-biased and MIM is texture-biased. In sum, self-supervised models trained with CL and MIM learn the representations in different levels of detail.

**Which components play an important role? (Section 4)** Analyses of the importance of each CL and MIM layer demonstrate that the later layers in CL and early layers in MIM play a key role. We interpret this as a consistent observation since early layers are usually known to capture low-level features—e.g., local patterns, high-frequency signals, and texture information—and later layers capture global patterns, low-frequency signals, and shape information (Dosovitskiy et al., 2021; Raghu et al., 2021; d'Ascoli et al., 2021; Graham et al., 2021; Dai et al., 2021; Park & Kim, 2022b).

From the above analyses and insights, we find that CL and MIM can complement each other and show in Section 5 that even the simplest implementation, such as a linear combination of CL and MIM objectives, can take advantage of both methods. Surprisingly, the hybrid models outperform those pre-trained with either CL or MIM both in terms of fine-tuning and linear probing accuracy.

## 2 HOW DO SELF-ATTENTIONS BEHAVE?

We point out that CL and MIM may not be silver bullets for all tasks, as shown in Figure 1. CL generally outperforms MIM in linear probing, while MIM dominates CL in the fine-tuning scheme. However, when we dissect the size of the model, CL outperforms MIM after fine-tuning for small models (cf. (Wang et al., 2022)), while MIM performs better on large models. Also, MIM yields effective representations for dense prediction tasks, such as object detection, but CL falls short on those tasks. This section explains these phenomena by investigating the behavior of self-attentions.

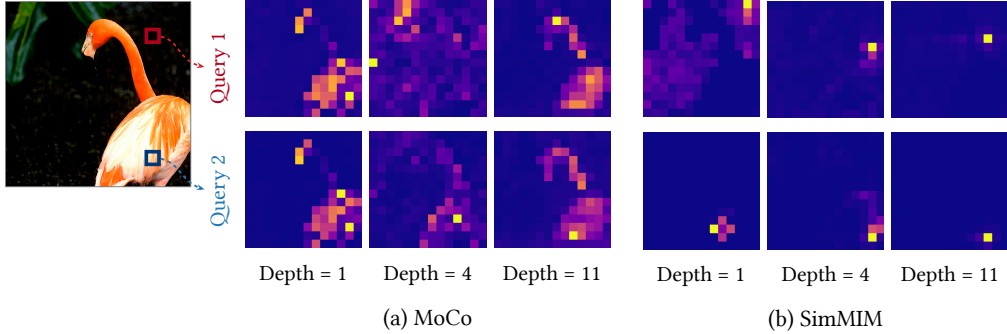

Depth = 1   Depth = 4   Depth = 11          Depth = 1   Depth = 4   Depth = 11

(a) MoCo                          (b) SimMIM

Figure 2: **Self-attentions of CL (MoCo) capture global relationships, but they collapse into homogeneous attention maps for all query tokens and heads.** Self-attentions of MIM (SimMIM) mainly focus on local areas. We visualize the attention maps for two different query tokens in the beginning through the end layers. We omit the results for self-attention heads, which show mostly consistent results. *Left*: Self-attentions of CL capture global patterns and the shape of an object. However, all attention maps capture the same shape information regardless of the query tokens. *Right*: Self-attentions of MIM capture local patterns and are correlated with query tokens.

Our analyses mainly compare ViT-B/16 pre-trained on ImageNet-1K (Russakovsky et al., 2015) with MoCo v3 (Chen et al., 2021) and SimMIM (Xie et al., 2022b). We use the ImageNet validation images for our experiments. We observe that other methods, e.g., DINO (Caron et al., 2021), BEiT (Bao et al., 2022), and MAE (He et al., 2022), have consistent properties (See Figure C.1).

**CL mainly captures global relationships.** We measure the ranges of self-attentions via attention distance (Dosovitskiy et al., 2021). Attention distance is defined as the average distance between the query tokens and key tokens considering their self-attention weights. Therefore, it conceptually corresponds to the size of the receptive fields in CNNs.

Figure 3 shows that the attention distance of CL (MoCo) is significantly higher than that of MIM (SimMIM), especially in the later layers. As seen in Figure 2, the qualitative visualization, this implies that the representations of CL contain global patterns and shape information, so CL can help ViTs distinguish between objects of images. Conversely, the self-attentions of MIM mainly capture local relationships; i.e., MIM may have difficulty recognizing whole objects and their shapes. Section 3 also discuss this claim from a representational perspective.

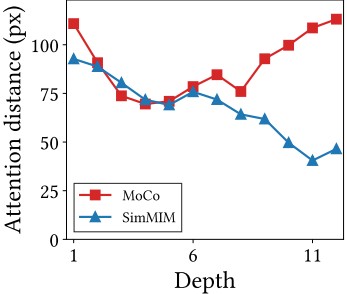

Figure 3: **Effective receptive fields of CL are global, but those of MIM are local.** This is particularly evident in the later layers.

**Self-attentions of CL collapse into homogeneity.** We observe an interesting behavior of CL in Figure 2, which shows the attention maps for query tokens from two different spatial locations. The self-attentions of CL surprisingly indicate *almost identical* object shapes for the two query tokens, compared to that of MIM. We describe this phenomenon as *an attention collapse into homogeneity*. This collapsing trend in the self-attentions of CL is observed across all the heads and query tokens. In contrast, the self-attentions of MIM are more faithful to the two query tokens, as expected.

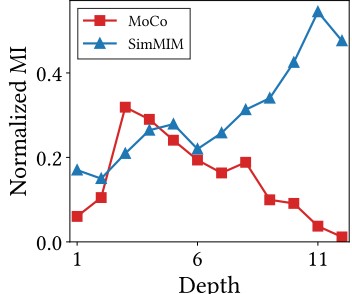

Figure 4: **Self-attentions of CL have little to do with query tokens.** Normalized MI of CL is significantly lower than that of MIM in the later layers.

We use normalized mutual information (NMI) (Strehl & Ghosh, 2002) to measure the attention collapse. Let $p(q)$ be a distribution of query tokens, and assume that these query tokens are uniformly distributed since a single query token is given for each spatial coordinate, i.e., $p(q) = 1/N$ where $N$ is the number of the tokens. Then the joint distribution of query and key tokens is $p(q, k) = \pi(k|q)p(q)$ where

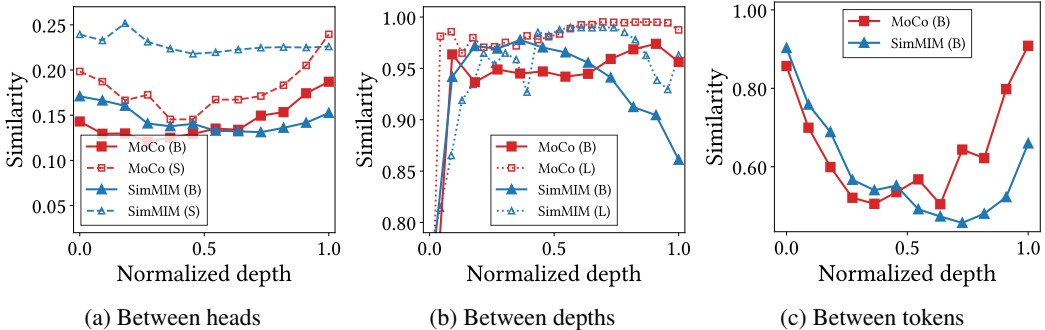

(a) Between heads  (b) Between depths  (c) Between tokens

Figure 5: **CL lacks representational diversity in the later layers.** We measure cosine similarities of representations in the self-attentions between the heads (left), depths (middle), and spatial coordinates (right). All of the results show that the representational similarity of later self-attentions of CL is higher than that of MIM. Increasing heads or depths of CL is not effective in improving the diversity. *Left*: The similarity of representations from two heads in self-attention. *Middle*: The similarity between representations before and after self-attentions transform them. *Right*: The similarities of representations at two spatial coordinates. ViT-{S, L} is trained with 100 epochs.

$\pi(k|q)$ is the softmax-normalized self-attention matrix. Thus, the normalized mutual information is $\frac{I(q,k)}{\sqrt{H(q)H(k)}}$ where $I(\cdot,\cdot)$ is the mutual information and $H(\cdot)$ is the marginal entropy. Low mutual information values show that attention maps are less dependent on the query tokens, implying an attention collapse into homogeneity. Conversely, high mutual information means that the attention maps strongly depend on the query tokens.

Figure 4 shows the degree of attention collapse in terms of the normalized mutual information (NMI). Results show that the mutual information of CL is significantly lower than that of MIM in the later layers, suggesting that the self-attentions of CL tend to collapse into homogeneous distributions.

**Attention collapse reduces representational diversity.** We conjecture that the self-attention collapse into homogeneity eventually leads to homogeneous token representations. To support this argument, we measure representational cosine similarities. In particular, we design three similarities: between different self-attention heads (heads), between the before and after self-attention layers (depths), and between different tokens (tokens).

Figure 5 shows the results, reporting the representation similarities for heads, depths, and tokens. As expected, the similarities of CL are notably higher than those of MIM in the later layers, indicating that the representations of CL have significant homogeneity. Even increasing the model size does not solve the problem CL has and may rather worsen it. Increasing the number of heads (ViT-S to ViT-B; Figure 5a) improves the representational diversity of MIM, but hardly improves the diversity of CL. Increasing the depth of CL (ViT-B to ViT-L; Figure 5b) only adds redundant modules.

**Implications of the behaviors we observed.** In conclusion, the self-attention of CL captures global patterns and shapes of objects. However, CL suffers from the problem of attention collapse into homogeneity, which reduces the diversity of token representations. On the other hand, MIM primarily captures local patterns and thus does not suffer from the attention collapse problem.

The behaviors mentioned above can explain the phenomena we observed in Figure 1:

- CL outperforms MIM in linear probing tasks because it captures shapes, which helps recognize objects and distinguish images. Although MIM preserves the texture and diversity of representations, their correlation with objects or content may not be as strong as shapes do.
- The attention collapse prohibits CL from fully exploiting heads, depths, and tokens of ViTs. Since homogeneous representations are not very helpful in improving token representations, ViTs trained with CL waste a large part of network capability. Therefore, the fine-tuning accuracy of MIM is significantly higher than CL in large models.
- CL is not suitable for dense prediction since the token features are homogeneous with respect to their spatial coordinates.

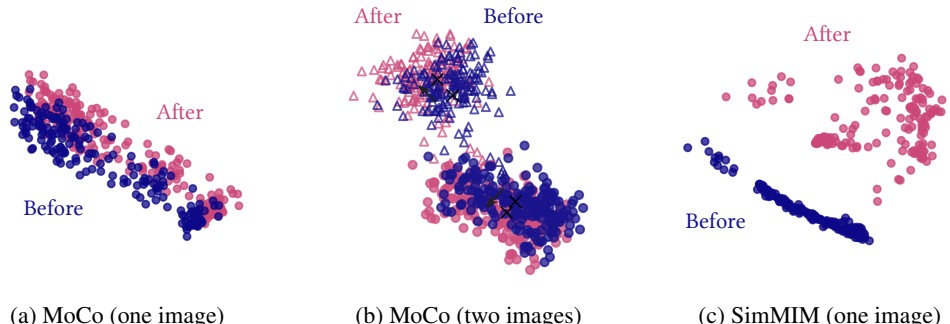

(a) MoCo (one image)   (b) MoCo (two images)   (c) SimMIM (one image)

Figure 6: **Self-attention layers of CL and MIM transform representations differently.** We visualize 196 spatial representation tokens for an example validation image in a representation space. The blue (●) and red (●) data points denote the tokens before and after the self-attention transformation. *Left*: The self-attentions of CL (e.g., MoCo) translate all the tokens equally, so the distances between the tokens of an image do not increase. *Middle*: However, CL moves the "centers of representations (represented by ×)" away from each other. Therefore, the images are linearly separable. The circle (●) and triangle (△) data represent tokens from different images. *Right*: The self-attentions of MIM (e.g., SimMIM) transform representations differently according to query tokens, thus increasing the distances between tokens. See Figure 7 for quantitative analyses.

We further investigate the self-attention's behavior with restricted receptive fields in Figure D.1. As shown in the experiment, locally restricted self-attentions lead to lower linear probing but higher fine-tuning accuracy, which is consistent with our observations.

## 3 How Are Representations Transformed?

In this section, we analyze the token representations of ViTs pre-trained with CL and MIM to demonstrate how the properties of self-attentions we observed in Section 2 affect the representations differently. We use the same pre-trained ViT-B/16 models by default default just as we did in Section 2.

**CL transforms all tokens in unison, while MIM does so individually.** To show how CL and MIM transform token representations, we visualize them in representation space. Figure 6 shows 196 (14×14 patches) tokens before and after self-attention modules from a single image sample of the ImageNet validation set. We use the three large singular vectors obtained via singular value decomposition (SVD) as the bases of the space. To better visualize this, we display the representation of MoCo and SimMIM in their crucial layers—the last layer and the first layer, respectively.

Figure 6a visualizes the changes that occur in the tokens of CL when transformed by self-attention module; it indicates that the self-attentions of CL translate all tokens in unison. This phenomenon occurs because the self-attention maps of CL are homogeneous, i.e., self-attention is almost independent of the spatial coordinates and query tokens. Therefore, the modules add near-constant to all the token representations. As a result, the inter-representation distance and the volume of representations do not increase, which implies that CL cares less about individual tokens.

Nevertheless, self-attentions are essential for the discriminative power of CL. As shown in Figure 6b, they help distinguish images by moving "the centers of the representation distribution" away from each other. In short, this figure suggests that CL makes the image linearly separable even though it loses the ability to distinguish tokens.

In contrast, MIM applies a different transformation to individual tokens, as shown in Figure 6c, because different self-attentions are assigned to the individual spatial tokens. Thus, MIM alters the distance between tokens of a single image as well as the volume of the representation distribution.

We find consistent results in quantitative analysis. Inspired by Jing et al. (2022), Figure 7 visualizes singular value spectra for tokens and images. A singular value spectrum provides singular values of a representation distribution obtained by SVD, so we can use it to represent the effective volume of distributions in a representation space. The higher the singular value in a spectrum, the larger the

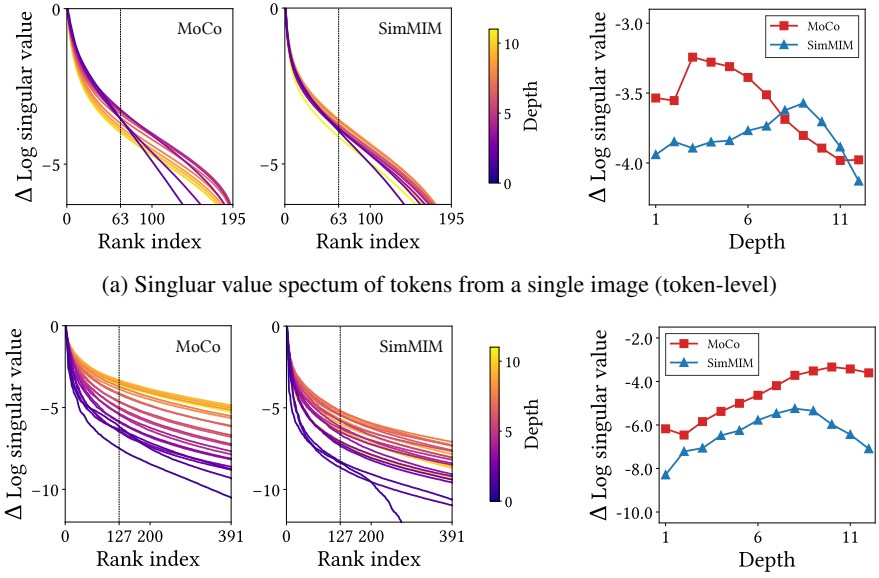

(a) Singluar value spectum of tokens from a single image (token-level)

(b) Singluar value spectum of images (image-level)

Figure 7: **CL barely changes or even decreases the distribution volume of tokens from a single image,** implying that it hardly distinguishes between token. Instead, it significantly increases the distribution volume of images. To demonstrate these properties, we visualize singular value spectra, the singular values of the distribution of representations sorted by the magnitude. The higher a singular value, the larger the volume of a distribution. The right of this figure shows the $64^{th}$ and $128^{th}$ highest singular value for depth. *Top*: Singular value spectra of tokens from a single image. CL decreases the singular values of the tokens, but MIM increases. *Bottom*: Singular value spectra of images. CL significantly increases the volumes occupied by images, but MIM hardly does so.

volume of a representation distribution. To calibrate the scale, we use the relative log singular value ($\Delta$ Log singular value), the difference with the (second) largest singular value for a depth.

Figure 7a shows singular value spectra of tokens from a single image. We calculate them for each image in the ImageNet validation set and report averaged singular values over the dataset. In this figure, the CL layers hardly increase or even decrease the singular value; consistent with the explanation above, this implies that CL hardly distinguishes tokens. In contrast, MIM increases the singular value, meaning that it changes the volume of tokens and can distinguish tokens. Another interesting observation is that a few later layers of MIM decrease the volume, even though they capture local patterns as shown in Figures 3 and 4. This is because they behave like decoders. Section 4 discusses this in detail.

Figure 7b shows the singular value spectra of images. We average all tokens in an image to build an image-level representation vector and conduct a singular value spectrum over the collection of representations in the validation set. As opposed to the previous case, the representational volume of CL is larger than that of MIM, which implies that CL makes the image-level representation separable.

**CL exploits low-frequencies, and MIM exploits high-frequencies.** We hypothesize that CL captures low-frequency and MIM captures high-frequency information in spatial dimensions since CL provides image-level self-supervision to capture global patterns, while MIM provides token-level self-supervision to exploit local patterns. To support this argument from a frequency perspective, we conduct a Fourier analysis of the representations as following Park & Kim (2022b). In particular, we report the relative log amplitude of Fourier-transformed representations by calculating the amplitude difference between the highest and lowest frequencies of representations.

Figure 9 visualizes the relative amplitudes of CL and MIM. It shows that the high-frequency amplitude of CL is significantly smaller than that of MIM, suggesting that CL mainly utilizes low-frequency spatial information such as global structures and shapes. On the contrary, MIM usually uses high-frequency spatial information such as narrow structures and fine textures.

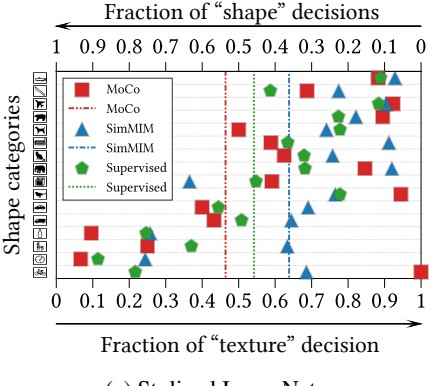
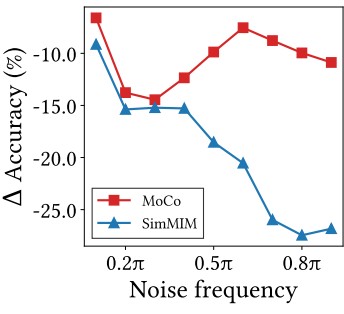

(a) Stylized ImageNet

(b) Robustness for noise frequency

Figure 8: **CL is biased toward shape, whereas MIM is biased toward texture.** We report the predictive results of models for linear probing tasks. However, we observe consistent results in fine-tuned models (See Figure F.2). *Left:* Result of classification on Stylized ImageNet. It shows that CL is more shape-biased than MIM and even than the supervised pre-trained model. Vertical lines represent averaged results for the shape categories. We also report the results of supervised ViT with ImageNet-1K class labels for comparison. *Right:* Accuracy drops on images with frequency-based random noises. MIM shows a more significant amount of accuracy drop than CL with high-frequency noises, demonstrating MIM's texture-biased property. The frequency window size of the frequency-based noise is $0.1\pi$.

Another interesting finding is that the last few layers of MIM reduce the high frequencies even though they only focus on local areas (See Figure 3). We conjecture that MIM implicitly divides ViTs into the encoder-decoder structure and allows intermediate layers to have linearly separable information. In contrast, CL allows the last layer to have such information. This is further elaborated in Figure 11.

**CL is shape-biased, but MIM is texture-biased.** Based on the results of the Fourier analysis, we assume that CL and MIM each have a bias toward shapes and textures, respectively. To demonstrate this claim, we use Stylized ImageNet (Geirhos et al., 2019), a texture-altered dataset, by using AdaIN (Huang & Belongie, 2017). Figure 8a reports the linear probing results on Stylized ImageNet to evaluate the shape and texture biases of pre-trained models. Compared to the model pre-trained with supervised learning, CL depends more on the shape and MIM depends on texture of images to classify images. In other words, CL is robust to texture changes, and MIM is vulnerable to them.

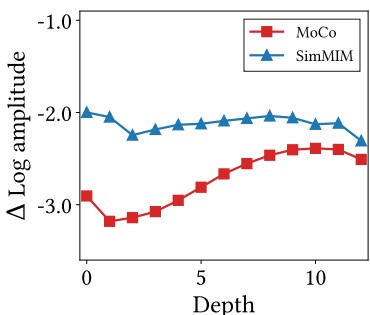

Figure 9: **CL exploits low-frequency, but MIM exploits high-frequency.** Moreover, a few last layers of CL reduce high-frequency by capturing global patterns. MIM also reduces it even though they capture local patterns, because the later layers behave like decoders. See also Figure 11.

Figure 8b shows the consistent results. In this experiment, we follow Park & Kim (2022a;b) and measure the decrease in accuracy on the ImageNet dataset with frequency-based random noise. The results suggest that CL is robust to high-frequency noises, but MIM is significantly more vulnerable to them. Since high-frequency noises harm the fine details of images, we arrive at the same conclusion that CL is more shape-biased and MIM is texture-biased. This can explain the robustness of CL against adversarial perturbations (Bordes et al., 2022).

## 4 WHICH COMPONENTS PLAY AN IMPORTANT ROLE?

The previous sections consistently show through various perspectives that CL exploits image-level global patterns while MIM captures token-level local patterns. This section analyzes pre-trained ViTs from an architectural perspective and shows that the key components in CL and MIM are different.

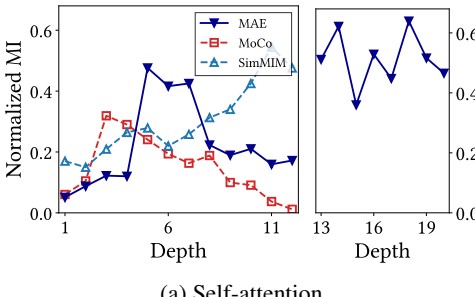 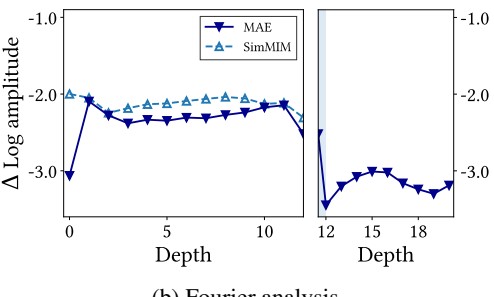

(a) Self-attention             (b) Fourier analysis

Figure 10: **The explicit decoder architecture of MAE helps ViTs effectively leverage the advantages of MIM.** We analyze the encoder and decoder of a pre-trained model with a masking ratio of zero. The left side of each figure represents the encoder and the right side the decoder. *Left:* The mutual information of MAE is lower than that of SimMIM in the encoder but higher in the decoder. *Right:* The decoder of MAE captures low-frequency information, and its encoder captures high-frequency information. Moreover, the later layers (excluding the last layer) of MAE do not reduce high-frequency information, while those of SimMIM do.

**Later layers of CL and early layers of MIM are important.** According to studies on ViT (Graham et al., 2021; Dai et al., 2021; Park & Kim, 2022b), the later layers use high-level information, and the early layers exploit low-level information. Since CL and MIM each exploit global and local patterns, we expect that the later layers of CL and early layers of MIM play a key role.

To evaluate the importance of each layer, we measure the linear probing accuracy using intermediate representations with the configuration of Table A.1. In Figure 11, we observe the following properties: First, the linear probing accuracy of MIM is higher than that of CL at the beginning. Conversely, CL outperforms MIM at the end of the model. Such result indicates that the later layers of CL and early layers of MIM play an important role in making linearly separable representations. Second, the accuracy of CL increases with increasing depth as expected, but the accuracy of MIM surprisingly decreases at the end of the model, i.e., the later layers of MIM are not very helpful in separating representations. We explain this observation as a phenomenon in which MIM methods with shallow prediction heads, e.g., SimMIM, use later layers of the backbone as a decoder. Therefore, MIM with a deep self-attention decoder, e.g., MAE (He et al., 2022), can be useful for linear probing performance. Moreover, it also explains why SimMIM's high-frequency component and representational volumes drop in the later layers as shown in Figures 7 and 9. Third, even the highest linear probing accuracy of MIM is lower than that of CL.

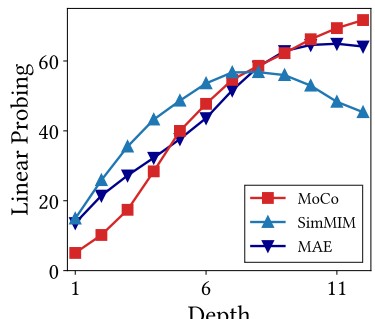

Figure 11: **Later layers of CL and early layers of MIM play a key role.** We report linear probing accuracies by using the representations of the intermediate layers. CL outperforms MIM in later layers, and MIM outperforms CL in early layers.

**The explicit decoder helps ViTs further leverage the advantages of MIM.** Several previous observations find that the implicit decoder of MIM with a shallow prediction head, such as SimMIM, can impair performance. MAE (He et al., 2022) addresses this problem by introducing deep explicit ViT decoders and reconstructing masked tokens only in the separate decoders.

In Figure 10, we analyze MAE to understand the properties of decoders more deeply. Figure 10a shows the self-attention behaviors. The results indicate that the mutual information of MAE is lower than that of SimMIM in the later layers of the encoder but higher in the decoder, implying that the decoder reconstructs masked tokens based on its neighborhood tokens.

Figure 10b shows the results of the Fourier analysis. As explained in Figure 9, the last four layers of SimMIM reduce the high-frequency components. In contrast, the later layers (excluding the last layer) of MAE do not reduce them. Instead, the decoder of MAE prioritizes low-frequency information compared with the encoder, allowing the backbone to efficiently utilize high-frequency information.

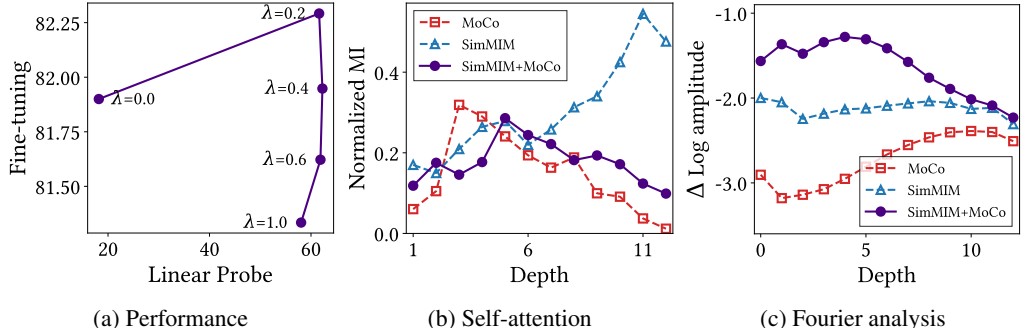

(a) Performance          (b) Self-attention          (c) Fourier analysis

Figure 12: **The simple linear combination of CL (MoCo) and MIM (SimMIM) objectives out-performs the vanilla CL and MIM**. $\lambda$ is the importance weight of CL, so $\lambda = 0$ means SimMIM and $\lambda = 1$ means MoCo. *Left:* "CL + MIM" outperforms CL and MIM in both linear probing and fine-tuning accuracy. *Middle:* Mutual information of "CL + MIM" decreases at the end of the model, suggesting that the self-attentions of later layers collapse into homogeneity and capture the same object shape information. *Right:* Fourier analysis shows that "CL + MIM" amplifies high frequencies at the beginning and reduces them at the end. It implies that "CL + MIM" exploits high-frequency information at the beginning and low-frequency information at the end.

## 5 ARE THE TWO METHODS COMPLEMENTARY TO EACH OTHER?

We present comparative analyses on CL and MIM from three perspectives: self-attentions, representation transforms, and the position of important layers. All of our results indicate that CL and MIM train ViTs differently. These differences naturally imply that combining CL and MIM to train a backbone may help leverage the advantages of both methods.

To show that CL and MIM are complementary, we introduce the simplest way to harmonize CL and MIM by linearly combining two losses, i.e., $\mathcal{L} = (1 - \lambda)\mathcal{L}_{\text{MIM}} + \lambda\mathcal{L}_{\text{CL}}$ where $\mathcal{L}_{\text{MIM}}$ and $\mathcal{L}_{\text{CL}}$ each indicate the losses of MIM and CL, and $\lambda$ is the importance weight of CL. We find that this simple hybrid model trained with combined losses efficiently exploits the strengths of both methods. Figure 12a shows linear probing and fine-tuning accuracy on ImageNet with varying $\lambda$. Surprisingly, the hybrid models outperform MIM ($\lambda = 0$) and CL ($\lambda = 1$) in both aspects. Figure 12b and Figure 12c can provide insights on how hybrid models behave by analyzing the model with $\lambda = 0.2$ in terms of self-attentions in Section 2 and Fourier analysis in Section 3, respectively; both results show that the hybrid model exploits MIM properties in the early layers and CL properties in the later layers. In particular, Figure 12b indicates that the self-attentions of the early layers are changed according to the query token but those of the later layers are not. Likewise, Figure 12c shows that the early layers exploit high-frequency, while the later layers try to exploit low-frequency.

## 6 CONCLUSION

We conducted a comparative study highlighting various facets of two widely used self-supervised learning methods for vision transformers: contrastive learning (CL) and masked image modeling (MIM). The study demonstrated many opposing properties of the two methods: image information (image-level *vs.* token-level; as in Section 2), feature representations (low-frequency *vs.* high-frequency; as in Section 3), and lead role components (later layers *vs.* early layers; as in Section 4). Furthermore, we suggested a possible application that exploits only the benefits from both methods and showed how a combined model can outperform individual methods.

**Future directions.** Various future directions can be explored based on our study. We believe that there are better ways than a simple linear combination of CL and MIM objectives. For example, a novel self-supervised learning approach, in which CL is applied in the later layers and MIM in the early layers, can be considered. Moreover, we may extend our findings on self-supervision for multi-stage ViTs, such as PiT (Heo et al., 2021) and Swin (Liu et al., 2021). Another interesting direction is to enhance the individual properties of CL and MIM. Techniques that help CL or MIM learn shapes or textures, respectively, may also improve performance.

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

## A  SETUP

We build the configurations based on Xie et al. (2022b) for fine-tuning and Caron et al. (2021) for linear probing. Table A.1 summarizes the configurations. Most analyzes of ViTs use the official ViT-B pre-trained models, and some analyzes use ViT-{S, L} pre-trained with official configurations but epochs of 100. Due to memory limitations, ViT-L is pre-trained with a quarter batch size of the other models. The hybrid model introduced in Section 5 uses the ViT backbone architecture of Xie et al. (2022b) and employs a configureation based on their work for pre-training as shown in Table A.1. For data augmentation and regularization, we adopt widely used settings, e.g., Randaugment (Cubuk et al., 2020), label smoothing (Szegedy et al., 2016), mixup (Zhang et al., 2018), cutmix (Yun et al., 2019), stochastic depth (Huang et al., 2016). Layer decay (Bao et al., 2022) is also used for fine-tuning. Neural network models are implemented in PyTorch (Paszke et al., 2019). The code for analysis is available at `https://github.com/naver-ai/cl-vs-mim`. All experiments use {1, 4, 8} NVIDIA A100 Tensor Core GPU. NSML (Kim et al., 2018) has been used for experiments.

Table A.1: **Training settings.**

| CONFIGURATION | Linear Probing | Fine-tuning | Pre-training |
|---|---|---|---|
| `optimizer` | sgd | adamw | adamw |
| `base learning rate` | 1.0e-0 | 1.25e-3 | 1.0e-4 |
| `weight decay` | 0.05 | 0.05 | 0.05 |
| `batch size` | 1k | 2k | 1k |
| `training epoch` | 50 | 100 | 100 |
| `learning rate schedule` | cosine | cosine | multistep |
| `warmup epoch` | 0 | 20 | 10 |
| `warmup schedule` | · | linear | linear |
| `randaugment` | · | 9, 0.5 | 9, 0.5 |
| `label smoothing` | · | 0.1 | 0.1 |
| `mixup` | · | 0.8 | 0.8 |
| `cutmix` | · | 1.0 | 1.0 |
| `stochastic depth` | · | 0.1 | 0.1 |
| `layer decay` | · | 0.65 | 1.0 |
| `gradient clip` | · | 5.0 | 5.0 |

## B  RELATED WORK

CL is a method based on comparing the global projection of two different random views. However, this approach usually suffers from the collapsing problem, where all representations collapse into constant solutions. To solve this problem, various methods such as negative samples and InfoNCE (Oord et al., 2018) have been proposed. Negative samples is an effective technique to avoid the collapsing problems, but they cause dimensional collapse (Jing et al., 2022) and require extra large batches (Chen et al., 2020a) or memory queues (He et al., 2020; Chen et al., 2020b) to retrieve them. We mainly analyze MoCo v3 (Chen et al., 2021), since the method includes these de facto standard components—global projection, random views, and negative samples.

Some SSL methods, e.g. Grill et al. (2020); Caron et al. (2021), do not use negative samples and use the projections of their Siamese representations as the positives. Such self-distillation has been explored theoretically and empirically (Chen & He, 2021; Tian et al., 2021) to prevent the collapsing problem, but we do not discuss the distillation scheme in this paper. Wei et al. (2022b) shows that feature distillation improves the fine-tuning performance of CL by diversifying attention ranges; this observation is consistent with our findings. While they focus on distillation to improve CL, we reveal the fundamental nature of self-supervised learning by rigorously comparing CL and MIM.

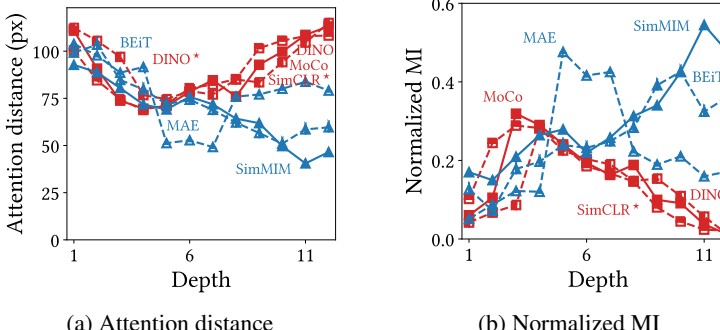

(a) Attention distance

(b) Normalized MI

Figure C.1: **MIM and CL methods each have consistent properties.** To show this, we visualize self-attention behaviors in terms of attention distance and normalized mutual information (MI). SimCLR⋆, which was introduced in Chen et al. (2021), stands for MoCo with a momentum coefficient of 0. *Left:* The attention distance of CL methods (namely MoCo, SimCLR⋆, and DINO) is higher than that of MIM methods (namely SimMIM, BEiT, and MAE). This suggests that CL methods consistently capture global patterns. *Right:* The normalized mutual information of MIM is higher than that of CL; i.e., the self-attentions of MIM are more correlated with query tokens than CL.

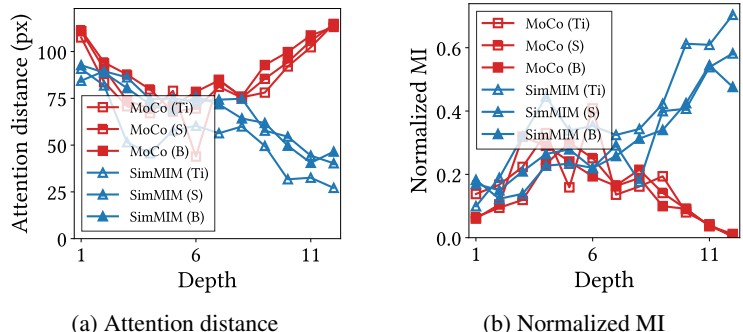

(a) Attention distance

(b) Normalized MI

Figure C.2: **ViTs exhibit consistent self-attention patterns, regardless of their size.** To better understand these patterns, we visualize the self-attention behaviors of three ViTs—ViT-{Ti, S, B}—using two metrics: attention distance and normalized mutual information (MI). *Left:* All self-attentions of MoCo capture global patterns in the later layers. In contrast, the self-attentions of SimMIM capture local patterns. *Right:* Likewise, all self-attention maps of MoCo collapse into homogeneity in the later layers.

Compared with CL, MIM has been rarely explored in vision tasks. Various methods, such as histograms of oriented gradients (Wei et al., 2022a) and tokenization (Bao et al., 2022), have been proposed as part of porting masked language models to the image domain with ViTs. Among them, SimMIM (Xie et al., 2022b) and MAE (He et al., 2022) are simple yet effective methods to reconstruct masked tokens without complicated pretext tasks. Because of its simplicity and superior performance in downstream operations, MIM is attracting attention as a promising technique in image processing.

Nevertheless, we find hints suggesting that CL and MIM utilize different aspects of the data, making them complementary. For example, Zhou et al. (2022); Wang et al. (2021); Yu et al. (2022) achieve high predictive performance by harmonizing the image-level and the token-level self-supervised learning. Xie et al. (2022a) also observe that, unlike supervised pre-trained models or CL, self-attentions in SimMIM focus locally; this is a consistent result with our findings.

## C OUR INSIGHTS ARE GENERALIZABLE TO VARIOUS MODELS

In the main text, we analyze ViT-B pre-trained using MoCo and SimMIM. We observe consistent characteristics across various sizes of ViTs that have been pre-trained using other self-supervised learning methods. To support this claim, we delve into the properties of self-attentions.

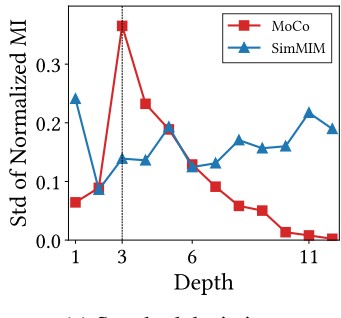
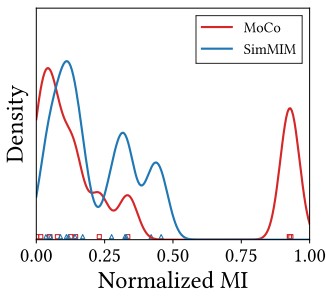

(a) Standard deviations

(b) Distribution at 3rd layer

Figure E.1: **The presence of an outlier head in MoCo raises the average of normalized mutual information.** This observation explains how the normalized mutual information in a couple of MoCo's self-attention layers is similar to or even surpasses that in SimMIM. *Left:* We present the standard deviation of the normalized mutual information. As depicted in this figure, the standard deviation in SimMIM remains relatively consistent across different depths. In contrast, the standard deviation in MoCo's 3rd or 4th self-attention layer is notably higher than that in SimMIM. *Right:* Distribution of mutual information for the third self-attention layer head. The visualization of this kernel density estimation shows that MoCo has an outlier head with mutual information close to 1.0. The red rectangles (□) and blue triangles (△) refer to the mutual information of heads in MoCo and SimMIM, respectively.

Figure C.1 visualizes the self-attention behaviors of different self-supervised learning methods in terms of attention distance and normalized mutual information. As depicted in the figure, all CLs and MIMs exhibit consistent properties. Similarly, Figure C.2 demonstrates that various sizes of models also demonstrate consistent properties.

## D  LOCALITY INDUCTIVE BIAS IMPROVES FINE-TUNING ACCURACY OF CL

In Section 2, we demonstrate that the homogeneity of self-attention map, i.e., attention collapse of CL, helps ViT distinguish images but harms fine-tuning accuracy. As a result, we anticipate that incorporating a locality inductive bias into CL will improve fine-tuning accuracy but degrade linear probing accuracy. One simple method to inject locality into self-attentions is to limit the receptive field of self-attention by using attention masks.

Figure D.1 shows the predictive performance of MoCo with restricted local self-attentions. As expected, the results are similar to the performance of MIM; As the kernel size decreases, the linear probing accuracy decreases but the fine-tuning accuracy increases. These results are consistent with our findings.

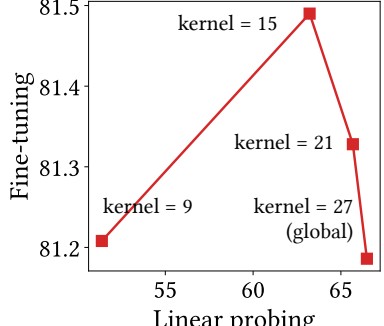

Figure D.1: **Locality inductive bias harms linear probing but improves fine-tuning.** We report the linear probing and fine-tuning accuracy of MoCo with restricted self-attentions via attention masks.

## E  A CLOSER LOOK AT THE ROLE OF SELF-SUPERVISED VIT LAYERS

The main text provides the key characteristics of CL and MIM. This section delves deeper into the details not covered in the main text to provide a more comprehensive understanding of the subjects.

**The role of the early modules.** Figures 3 and 4 suggest that most layers of MoCo capture global patterns and have only a weak correlation with query tokens. However, one or two of MoCo layers exhibit unusual behavior. For example, the 3rd layer of MoCo focuses on local areas and its self-attention map is dependent on the query. We explore this property in more detail.

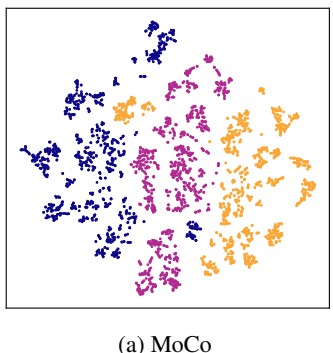
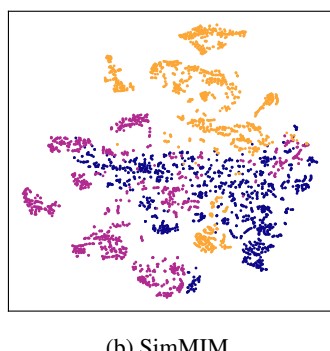

(a) MoCo          (b) SimMIM

Figure E.2: **The tokens of MoCo form a cluster for each image, while those of SimMIM are intermingled.** This aligns with the finding that, compared to SimMIM, MoCo is linearly separable. To demonstrate this property, we visualize 3,528 tokens (196 tokens×18 images) from the representations of the last layer via t-SNE, and find that a consistent pattern is observed even in the representations of the intermediate layers. The colors represent three different classes. See also Figures 6 and 7.

Figure E.1a provides the variance of normalized mutual information with respect of heads. As the results show, the variance of SimMIM is consistent across all depths whereas that of MoCo is not. In particular, the 3rd layer of MoCo has high variance even though other layers do not. This suggests that, while most of MoCo's self-attention heads capture global patterns and have weak correlation with query tokens, some heads deviate from this behavior and exhibit a different pattern.

Figure E.1b shows the distribution of normalized mutual information among heads in the 3rd layer to analyse this phenomenon. In this figure, we use kernel density estimation with Gaussian kernel to visualize the distribution. The results reveal several outlier heads in MoCo with mutual information close to 1.0. As a result, these outliers significantly raises the average value of normalized mutual information.

**A comprehensive view through visualization of tokens from multiple images.** Figure 6 visualizes how self-attention layers transform tokens from one or two images in representation space. The figure demonstrates that MoCo transforms all tokens in union while SimMIM transforms them individually. As a result, MoCo separates the representations at the image-level and SimMIM separates them at the token-level.

The t-SNE visualization (Van der Maaten & Hinton, 2008) in Figure E.2 provides consistent results and offers even a more comprehensive perspective. In this figure, we visualize the last representations of 3528 tokens from 18 images that belong to three different classes. The visualization demonstrates that MoCo separates the representations into distinct classes and even images, while maintaining the tokens close together in compact image clusters. On the other hand, SimMIM separates tokens from images, resulting in a wide representation space for each image, but the images or even classes may be challenging to linearly distinguish.

**The first layer of MoCo aggregates tokens into compact clusters.** Figures 6 and 7 shows that all modules, except the first module, in MoCo behave consistently. However, we observe that MoCo's first module behaves differently and unusually than the others. We elaborate the behaviour of the first layer of module.

Figure E.3a shows the qualitative visualization of tokens for a sample image, similar to Figure 6. This visualization shows that the first MoCo layer aggregates tokens into compact clusters. Although this figure only uses a single image, the layer aggregates all images into a small representation space as well. In terms of singular values, we observe consistent results. Similar to Figure 7, Figures E.3b and E.3c report the second largest log singular value, instead of the relative log singular value, to investigate the absolute volume of the representations. As expected, most layers in both MoCo and SimMIM increase the singular value, but surprisingly, the first layer of MoCo reduces the singular value, meaning that the volumes of representations are decreased at both the token-level and image-level. Based on these observations, we conjecture that the first module of MoCo behaves like an embedding component.

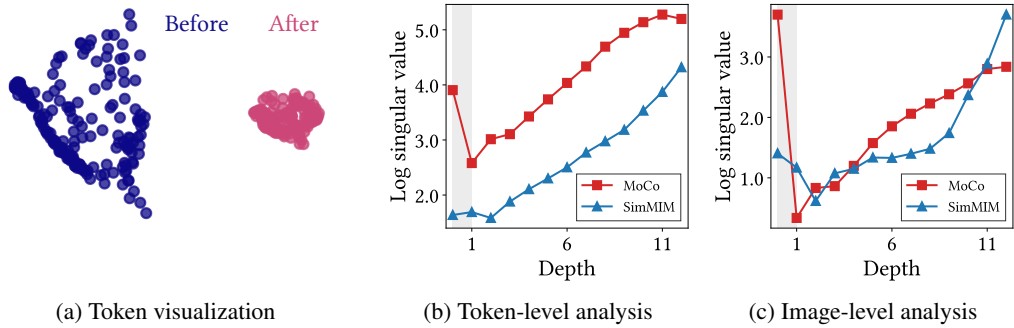

(a) Token visualization      (b) Token-level analysis      (c) Image-level analysis

Figure E.3: **The first layer of MoCo clumps tokens together.** We demonstrate this property from two perspectives: qualitative visualization and singular value of token distribution. *Left:* Similar to Figure 6, we visualize tokens of a sample image in a representation space. The blue and red data points represent the tokens before and after the self-attention transformation. As shown in this figure, the first self-attention layer clumps tokens into a compact cluster. *Middle and Right:* Similar to Figure 7, we visualize the second largest log singular value (not $\Delta$ log singular value) for depth. The singular value spectra demonstrate consistent results; the first layer of MoCo (gray area) not only clumps tokens but also images into a compact cluster.

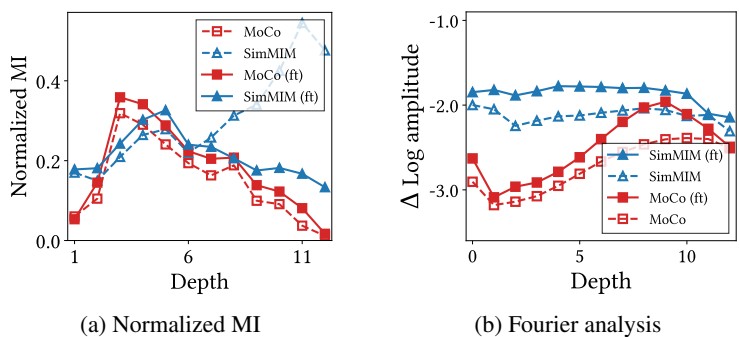

(a) Normalized MI      (b) Fourier analysis

Figure F.1: **Self-attentions and representations in fine-tuned models exhibit consistency with those of pre-trained models.** Similar to Figures 4 and 9, we present the normalized mutual information and the Fourier analysis results of fine-tuned models. The abbreviation "ft" stands for "fine-tuned model." *Left:* Similar to pre-trained models, the mutual information of MoCo's self-attention maps is generally lower compared to that of SimMIM. However, it is noteworthy that the mutual information of the later self-attention maps in SimMIM decreases significantly. This is because the later layers of a model trained with supervision or fine-tuning tend to capture global information. *Right:* Similarly, SimMIM utilizes higher frequency information than MoCo.

## F   Fine-Tuned Models Inherit the Properties of Pre-trained Models

The main text focuses on highlighting the key properties of pre-trained models. This section demonstrates that these properties are also utilized by fine-tuned models. As a result, we can safely apply the insights gained from the main text to various situations.

**Consistent results in self-attention and Fourier analysis.** Figures 3 and 4 demonstrate that MoCo captures global areas and that its self-attentions are less related to the query tokens, compared with SimMIM. In addition, Figure 9 shows that MoCo captures low-frequency information as opposite to SimMIM. These results are consistent in the fine-tuning scheme.

Figure F.1a reveals the self-attention behaviours of fine-tuned MoCo and SimMIM in terms of normalized mutual information. Similar to pre-trained models, the fine-tuned self-attention maps of

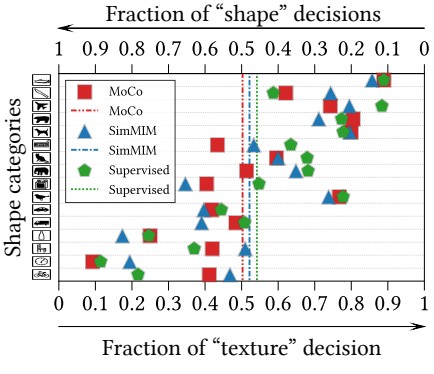

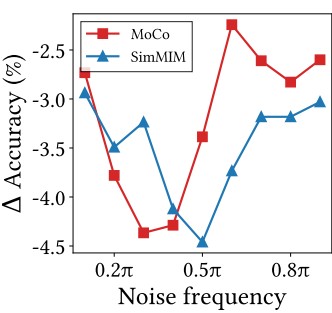

(a) Stylized ImageNet

(b) Robustness for noise frequency

Figure F.2: **Fine-tuned ViTs inherit the robustness against frequency-based noise.** Similar to Figure 8b, we measure the decrease in the accuracy of ViTs fine-tuned with MoCo and SimMIM. *Left:* Even with fine-tuned ViTs, MoCo is relatively shape-biased and SimMIM relatively texture-biased. This bias is just less apparent than in linear probing models. *Right:* The robustness against frequency-based random noise also suggests the same: MoCo is robust against high-frequency noise, but SimMIM is not. In conclusion, fine-tuned models inherit the properties of linear probing models.

MoCo have generally lower mutual information compared to those of SimMIM. The only significant difference is that the mutual information of the later self-attention maps in fine-tuned SimMIM decreases significantly, as later layers in models trained with supervision or fine-tuning tend to capture more global information. As a result, the gap between the two methods is reduced. This is also reflected in the consistent results of Fourier analysis as shown in Figure F.1b. In this analysis, SimMIM captures higher-frequency information compared to MoCo in fine-tuning scheme as well. However, the later layers of SimMIM attempt to capture low-frequency information. Therefore, the gap of fine-tuned models is smaller than that of pre-trained models.

**CL is shape-biased and MIM is texture-biased in fine-tuning scheme.** In Figure 8, we demonstrate that linear probing model with CL (MoCo) is more shape-biased and that with MIM (SimMIM) is texture-biased, compared with each other. As in the experiment, we calculate the classification results of ImageNet fine-tuned MoCo and SimMIM on Stylized-ImageNet, and measure the decrease in accuracy against frequency-based random noise. As we would expected, Figure F.2 shows that the property also extends to the fine-tuned model. Even though we still observe the difference between MoCo and SimMIM, the performance gap between MoCo and SimMIM is quite reduced compared to the gap between the linear probing models.

**Later layers of CL and early layers of MIM are important in find-tuning phases.** As shown in Figure 11, the later layers of the CL and the early layers of the MIM are linearly separable. This finding suggests that these layers are significant, however, it does not provide direct evidence that such properties are preserved during fine-tuning phases. We demonstrate that these layers play a crucial role in fine-tuning phases as well.

To support this claim, we conduct a study to measure the accuracy drop of fine-tuned models using pre-trained models with a few blocks initialized. As shown in Figure F.3a, the results indicate that the initializing a few early blocks in the pre-training models of SimMIM significantly harms the fine-tuning accuracy, compared to MoCo. These observations suggest that early layers of SimMIM play an important role in fine-tuning. Conversely, Figure F.3b shows that initializing later

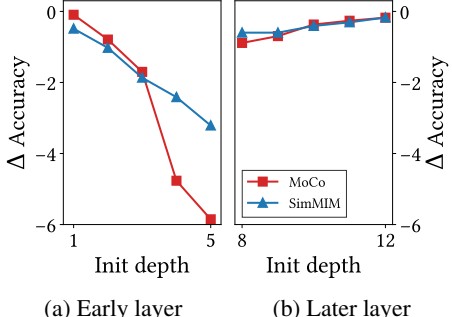

(a) Early layer     (b) Later layer

Figure F.3: **The later layers of CL and early layers of MIM play a key role in the fine-tuning scheme.** To show this, we initialize a few blocks and measure the decrease in the fine-tuning accuracy of pre-trained models.

blocks in the pre-training models of SimMIM does not significantly harms the fine-tuning accuracy, suggesting that they are not important in fine-tuning compared with MoCo.

One limitation of this experiment is the evaluation of the accuracy drop in a single run. Since the accuracy drop of MoCo is marginally higher than that of SimMIM at the first initialization depth in Figure F.3b, additional experiments may improve the results. In this experiment, we utilized the same fine-tuning settings for both MoCo and SimMIM; but experiments with fine-tuning settings tailored to each method may provide further insight.

## G    HYBRID MODELS OUTPERFORM CL AND MIM IN DOWNSTREAM TASKS

The claim that CL and MIM are complementary is demonstrated only on ImageNet in Section 5. To validate this claim in tasks beyond ImageNet, we evaluated the pre-trained models of the hybrid method introduced in Section 5 for another classification task and a semantic segmentation task. In particluar, we measured the accuracy on iNaturalist 2018 (Van Horn et al., 2018) and the mIoU on ADE20K (Zhou et al., 2019). As shown in Table G.1, the hybrid

Table G.1: **Hybrid models of CL and MIM outperform both CL and MIM in various tasks.**

| $\lambda$ (IMPORTANCE OF CL) | iNat-18 | ADE20k |
|---|---|---|
| 0.0 (SimMIM) | 62.1 | 35.4 |
| 0.2 (SimMIM + MoCo) | **68.8** | **42.2** |
| 1.0 (MoCo) | 66.2 | 39.7 |

model of SimMIM and MoCo outperforms both SimMIM and MoCo in various downstream tasks. Therefore, we conclude that the effectiveness of this claim extends beyond ImageNet.

