# OpenReview forum: "What Do Self-Supervised Vision Transformers Learn?"
_ICLR.cc/2023/Conference — ICLR 2023 poster_

### Official Review · Reviewer_Gfy8 · 2022-10-21

**Confidence:** 4
**Clarity, Quality, Novelty And Reproducibility:** Well-written, interesting conclusions…
**Correctness:** 2
**Technical Novelty And Significance:** 3
**Empirical Novelty And Significance:** 3
**Recommendation:** 6

**Strength And Weaknesses:**

Strength:
Well-written, clear analysis and easy to understand.
The conclusions drawn in the paper may be instructive to the self-supervised learning community.

Weaknesses:
(1) The main concern lies in the quantitative experiments. In fact, the paper lacks experiments on downstream tasks, like image classification, object detection or segmentation. The authors should provide corresponding downstream experiments to support the effectiveness of the conclusion other than visualization of principle analysis.
(2) The principle analysis occupies a large amount of space in the paper, on the contrary, the method derived from the conclusions is rarely involved. Actually, the corresponding method (Section 5) is also the core of the paper.


**Summary Of The Paper:**

The paper mainly investigates the principle differences between contrastive learning (CL) and masked image modeling (MIM). That’s the CL is shape-biased, which learns low-frequencies, and the MIM is texture-biased, which exploits high-frequencies. The Conclusion is interesting and the analysis is thorough.

**Summary Of The Review:**

I am inclined to reject the current version of the paper since the experiment and the corresponding method are not well explored.

---

> ### Author Response · Authors · 2022-11-19
> **Author Response to Reviewer Gfy8**
>
>
> We appreciate your feedback and suggestions. We have addressed your concerns below:
>
> ---
>
> **Ⅳ-1. The authors should provide corresponding downstream experiments to support the effectiveness of the conclusion other than visualization of principle analysis.**
>
> This paper demonstrates the claim of Section 5 – CL and MIM are complementary – only on ImageNet. As per your suggestion, we evaluate the pre-trained models of Section 5 on two additional datasets to further extend the effectiveness of the claim beyond ImageNet. As expected, the result in Table F.1 shows that the hybrid model (SimMIM+MoCo) outperforms SimMIM and MoCo in another classification downstream task (iNaturalist 18) and a semantic segmentation task (ADE20k).
>
> Due to the time constraint of the reviewer-author discussion period, we have not been able to perform experiments in various tasks. However, we will try to add the results on other classification datasets and COCO in the camera-ready version.
>
> ---
>
> **Ⅳ-2. The principle analysis occupies a large amount of space in the paper. On the contrary, (in Section 5,) the method derived from the conclusions is rarely involved.**
>
> *Regarding the large amount of space occupied by the analysis*: Although proposing a novel self-supervised method is important, we believe that analyzing self-supervised learning methods to reveal how they work is also an important and well-appreciated research goal in the community. In that respect, as mentioned by you and agreed by Reviewer 54TF and Reviewer wzZN, the key contribution of this paper is to present the interesting analysis in an easy-to-understand manner, not to propose a novel method. We hope that the insights we have provided will be instructive and useful in designing future self-supervised learning techniques.
>
> ·
>
> *Regarding Section 5*: To *analyze* the standard behavior of the hybrid model of CL and MIM, Section 5 introduces the simplest combination of these. The observation also shows that even the simplest harmonization can help leverage the advantages of both methods on ImageNet. We would like to clarify that our overriding goal is not to design a novel method that fully exploits the conclusions of the previous sections. These results may be helpful in understanding how hybrid models behave.
>
> ·
>
> Thank you for your valuable feedback to improve the paper. To avoid misunderstanding, we will revise Section 5 to make this clearer.

---

> > ### Comment · Reviewer_Gfy8 · 2022-12-07
> > **Comments after rebuttal**
> >
> > Thanks for the response. Overall, the rebuttal has dispelled my concerns. Please add the corresponding results and reorganize Section 5 in the revised manuscript.

---

### Official Review · Reviewer_n5hD · 2022-10-23

**Confidence:** 5
**Correctness:** 3
**Technical Novelty And Significance:** 1
**Empirical Novelty And Significance:** 1
**Recommendation:** 3

**Clarity, Quality, Novelty And Reproducibility:**

Originality of this paper: Weak and missing. This paper is only about the evaluation of two standard methods.

Quality of the research: Limited contributions towards the topic. The evidence to justify the argument may be further improved - more metrics and comprehensive evaluation to be undertaken.

Clarity of the paper: In general, the entire paper is easy to follow. The major concern is that the combination of the two methods is not well explained.

**Details Of Ethics Concerns:**

Datasets used in this study come from the publicly accessible databases so no concern on ethics.

**Strength And Weaknesses:**

Strengths of the this paper -

The entire paper, in its current form, is easy to follow. The examination of certain components in each method is clearly understandable. The evidence presented in the paper seems reasonable.

Weaknesses of the paper -

The major weakness is the lack of novelty and comprehensiveness. The comparisons show the examination of two methods, given a large number number  of similar approaches reported in the community.
Another weakness of the paper is that the mixture of the two methods is to simply use the weighted combination of the two methods. In spite of its simplicity, this combination form lacks clear motivation and clear discussion on the merging form, e.g. why this form is the best way, and how to merge them in the level of components?

Actions to be taken:

More similar techniques must be taken into account. The comparison aspects should be considered, e.g. components and layers of the systems. The integration of the systems should be properly reasoned.

**Summary Of The Paper:**

This paper presents comparative studies on various facets of two widely used self-supervised learning methods - contrastive learning and masked image modeling. The studies show opposing properties of the two methods: image information (image-level vs. token-level), features representations (low- and high-frequency) and different lead role components (later- and early-layers). The study also demonstrated that their misture outperforms individuals of them.

**Summary Of The Review:**

This paper presents the comparisons of two standard methods with several evaluation metrics. The strength of the paper is the easiness to understand the presented work. The major weaknesses of the paper include the missing depth and width of the comparisons. This paper and its presentation in the current form will not be recommended for acceptance.

---

> ### Author Response · Authors · 2022-11-19
> **Author Response to Reviewer n5hD**
>
>
> Thank you for your review and constructive feedback. Below, we address your concerns, and clarify the objectives of this paper and Section 5.
>
> ---
>
> **ⅠⅠⅠ-1. The major weakness is the lack of novelty and comprehensiveness.**
>
> *Regarding the novelty and originality*: As mentioned in L31-33 and as you correctly summarized, the overriding goal of this paper is to analyze the behaviour of ViTs trained through CL and MIM and show that they learn opposite knowledge. Although it is important to propose improved self-supervised learning methods, we believe that analyzing the behavior of self-supervised learning methods is also an important research topic that is highly appreciated by the community. For example, prior empirical analysis papers on neural networks [a] and ViTs [b,c,d] have made extensive and substantial contributions to the communities including ICLR by providing novel insights. We hope that the insights provided in our self-supervised analysis paper will be useful in designing future self-supervised learning methods, as the previous analysis work did.
>
> In short, our key contribution is to present an interesting (*Reviewer 54TF*, *Reviewer wzZN*, *Reviewer Gfy8*), original (*Reviewer 54TF*) analysis of CL and MIM in an easy-to-understand manner (*Reviewer 54TF*, *Reviewer wzZN*, *Reviewer n5hD*, *Reviewer Gfy8*). We would like to clarify that, as *Reviewer 54TF* correctly figured out, proposing a novel self-supervised method is *not* the key contribution and is beyond the scope of this paper. Please refer to ⅠⅠⅠ-2 below for a detailed discussion of Section 5 (linear combination of CL and MIM).
>
> ·
>
> *Regarding the comprehensiveness*: We believe that this paper provides a comprehensive analysis of CL and MIM as *Reviewer 54TF* and *Reviewer wzZN* commented. We provide quantitative as well as qualitative results, and claims are supported by various metrics. Moreover, as *Reviewer wzZN* pointed out, we do not stop at the first finding, but try to analyze them from various perspectives. As a result, this paper provides consistent but varied insights.
>
> To achieve this goal, we mainly compare MoCo and SimMIM. These two methods can be regareded as standard methods including only canonical techniques, and are suitable targets for highlighting the properties of CL and MIM. In general, many self-supervised methods use their own tricks to improve performance, so investigating all of them is not our primary goal.
>
> Nevertheless, based on your feedback, we added analysis of MAE [e] in Figure 10 to improve the discussion of the role of each component. Because of its wide impact and effectiveness, the deep explicit decoder trick introduced by MAE [e] is worth discussing in detail. The results show that the explicit decoder architecture helps ViT effectively leverage the advantages of MIM. As a result, the explicit decoder can be also useful for linear probing performance as shown in the additional experiment Figure 11. Moreover, we try to show that various MIM methods and CL methods each have consistent properties in Figure B.1.
>
> ·
>
> *In summary*, our contribution lies in the analysis rather than in proposing a novel technique. Although analyzing all self-supervised learning methods is not our overriding goal, we did our best to show that our observations may be generalized.
>
> Thank you for your valuable feedback. Based on your feedback, we were able to further improve the manuscript.
>
> ---
>
> **ⅠⅠⅠ-2. Why the weighted combination of CL and MIM is the best way?**
>
> We would like to clarify that Section 5 does *not* aim to propose a new self-supervised method, nor claim that the weighted combination of CL and MIM is the best way. Instead, Section 5 provides a deeper analysis of the behaviors of hybrid self-supervised ViTs by exploiting the most straightforward combination of CL and MIM. In this section, we empirically show that even the simplest harmonization can help leverage the advantages of both methods, supporting our analysis and claims in the previous sections, "CL and MIM work differently." These results may be helpful in understanding how hybrid models behave.
>
> We appreciate your constructive feedback to improve the manuscript. To avoid confusion, we will revise Section 5 to make this clearer.
>
> ·
>
> [a] Veit, Andreas, Michael J. Wilber, and Serge Belongie. "Residual networks behave like ensembles of relatively shallow networks." NIPS (2016). \
> [b] Raghu, Maithra, et al. "Do vision transformers see like convolutional neural networks?." NeurIPS (2021). \
> [c] Naseer, Muhammad Muzammal, et al. "Intriguing properties of vision transformers." NeurIPS (2021). \
> [d] Park, Namuk, and Songkuk Kim. "How Do Vision Transformers Work?." ICLR (2022). \
> [e] He, Kaiming, et al. "Masked autoencoders are scalable vision learners." CVPR. 2022.

---

> > ### Comment · Reviewer_n5hD · 2022-11-28
> > **further comments after rebuttal**
> >
> > Many thanks the authors for their continuous work and responses to the review comments. However, the rebuttal is not convincing as this conference is looking at insightful discussion on technology and/or new progress on algorithmic development but the current paper is not answering any of these calls. Therefore, the reviewer will keep the original review score.

---

### Official Review · Reviewer_wzZN · 2022-10-24

**Confidence:** 3
**Correctness:** 4
**Technical Novelty And Significance:** 3
**Empirical Novelty And Significance:** 3
**Recommendation:** 8

**Clarity, Quality, Novelty And Reproducibility:**

Quality: high. The core idea is relevant and the execution is well done through a series of well-discussed experiments.

Clarity: high. The paper is well-written and well-organized. The various sections are connected in a logical way.

Originality: low. No new idea is introduced, apart from combining contrastive learning and masked image modeling which can be found in other concurrent works. No new "tool" for probing the learned representations is introduced either. However, new tools and ideas are not always required and the paper shines for the quality of the execution and the discussion.

Reproducibility: high. Although the code is not yet available, it should be possible to fully reproduce the experiments in the paper based on the referenced methods.

**Strength And Weaknesses:**

Strengths:

- The paper tackles interesting questions in self-supervised learning and provides answers that are thoroughly supported by empirical evidence. With a possible risk of confirmation bias, the findings in this paper validate the anecdotal evidence that has been observed in the community for a while.

- From the reader's perspective, the paper is well-written and well-organized. The authors have done a great job of outlining the research questions and their findings in the introduction.

- The research presented in this work is quite extensive. I appreciate that the authors did not stop at the first finding and instead insisted on probing the models with different tools to highlight their differences from various perspectives.

Weaknesses (or rather suggestions for improvement):

- Complement the discussion by mentioning the findings of "High fidelity visualization of what your self-supervised representation knows about" (Bordes 2022). This work, concurrent with the authors' work, offers qualitative evidence that broadly agrees with the findings of this paper.

- L59-61 "early layers are usually known to capture low-level features, e.g., local patterns, high-frequency signals, and texture information, and later layers capture global patterns, low-frequency signals, and shape information". In the context of convolutional networks, due to their limited receptive field and pyramidal architecture, this statement is considered common knowledge and is supported by a large body of studies. However, this paper focuses on transformers, where this behavior is still under investigation. I suggest providing references to support this statement and possibly clarifying its scope.

- Most of the discussion points compare MoCo vs. SimMIM. However, there are parts that would be more convincing if more methods were added to the discussion. An example of this, L257-267 discuss the difference between SimMIM and MAE in terms of decoder depth, why not include MAE in figure 11?

Minor points:

- Improve figure 1 by adding a grid, separating the plots more, and tweaking the font size. Actually, most figures would benefit from a grid.

- L67: "As shown in Figure 1". Figure 1 deserves a bit of introduction, e.g. what models and methods are being compared and on what, before discussing its meaning. I see that the caption contains all the required information, but it is weird to read the first line of section 2 without knowing what the figure is about.

- Improve figure 10 by connecting all MLP markers with a line, and connecting all self-attention markers with another line, rather than relying on white/gray background to distinguish them. Alternatively, if you want to use a single line for each method, change at least the markers of MLP and self-attn layers.

**Summary Of The Paper:**

The paper compares two self-supervised classes of methods for vision transformers, namely contrastive learning and masked reconstruction.
Through several quantitative analyses of pre-trained models, the authors expose a series of findings on the learning properties of these methods.
The main findings for contrastive learning are: it captures global low-frequency patterns, its intermediate representations are rather homogeneous, and self-attention is its key component.
Conversely, for masked reconstruction, the main findings are: it focuses on local high-frequency patterns, its intermediate representations maintain diversity, and the MLPs are its key component.


**Summary Of The Review:**

This paper offers a quantitative study of various properties of contrastive learning and masked reconstruction in self-supervised vision transformers. Such a study is valuable as it validates what has been observed empirically in the community and provides a tentative analysis of the reasons behind these observations. The paper itself is well-written and clearly outlined. For these reasons, I recommend the paper for publication.

---

> ### Author Response · Authors · 2022-11-19
> **Author Response to Reviewer wzZN**
>
>
> Thank you for your encouraging and constructive feedback. We have addressed your concerns below.
>
> ---
>
> **ⅠⅠ-1. The paper [a], concurrently with the authors' work, offers qualitative evidence that broadly agrees with the findings of this paper.**
>
> Thank you for suggesting this related work. We agree that [a] provides qualitative results that agrees with our results. In particular, the finding of the robustness of CL methods against adversarial perturbations is consistent with our conclusions. Furthermore, we believe that it is promising to investigate what knowledge neural networks learn through the lens of generative models, and is also interesting future work to analyze MIM methods through such a lens.
>
> ---
>
> **ⅠⅠ-2. Provide references to support the statement, "early layers are usually known to capture low-level features, and later layers capture high-level features".**
>
> We have added some references to support the statement. For example, [b,c] observe that self-attentions capture local relationships in early layers and long-range relationships in later layers. [d,e,f] demonstrate that locality inductive bias at the early layers improves the performance of ViTs. [g] shows that early self-attentions amplify high-frequency information, and later layers reduce it.
>
> ---
>
> **ⅠⅠ-3. Why not include MAE in Figure 11 (linear probing experiment)?**
>
> Thank you for the suggestion. As per your suggestion, in Figure 11, we have added the linear probing result of MAE. The result shows that the linear probing accuracy of MAE is superior to that of CL at the beginning, but it is inferior at the end. More importantly, as we would expect, the accuracy of MAE does *not* decrease at the end of the model, unlike that of SimMIM. This result also support the claim that the explicit decoder of MAE help leverage the advantages of MIM. We believe that this additional experiment improves the paper.
>
> ---
>
> **Minor points. Improve Figure 1 / it is weird to read the first line of Section 2 without knowing what Figure 1 is about / Improve Figure 10.**
>
> Thank you for your detailed suggestions. We agree that these visualizations can be improved. During this discussion period, we will continue to do our best to improve the manuscript, and the the camera-ready version will be refined taking this into account.
>
>
> ·
>
> [a] Bordes, Florian, Randall Balestriero, and Pascal Vincent. "High Fidelity Visualization of What Your Self-Supervised Representation Knows About." arXiv preprint arXiv:2112.09164 (2021). \
> [b] Dosovitskiy, Alexey, et al. "An image is worth 16x16 words: Transformers for image recognition at scale." ICLR (2021). \
> [c] Raghu, Maithra, et al. "Do vision transformers see like convolutional neural networks?." NeurIPS (2021). \
> [d] d’Ascoli, Stéphane, et al. "Convit: Improving vision transformers with soft convolutional inductive biases." ICML (2021). \
> [e] Graham, Benjamin, et al. "Levit: a vision transformer in convnet's clothing for faster inference." ICCV (2021). \
> [f] Dai, Zihang, et al. "Coatnet: Marrying convolution and attention for all data sizes." NeurIPS (2021). \
> [g] Park, Namuk, and Songkuk Kim. "How Do Vision Transformers Work?." ICLR (2022).

---

### Official Review · Reviewer_54TF · 2022-10-26

**Confidence:** 4
**Correctness:** 3
**Technical Novelty And Significance:** 2
**Empirical Novelty And Significance:** 3
**Recommendation:** 6

**Clarity, Quality, Novelty And Reproducibility:**

**Quality & Clarity**: the paper is in general well written. The figures are neat with detailed captions.

**Novelty**: the analyses and findings are largely original and interesting, though the analysis tools used in the paper are borrowed from other works.

**Reproducibility**: Though the code for pre-training and fine-tuning the models are provided in supplementary material, the code to reproduce the figures (or analysis results) is absent, which is the key of this work.

**Strength And Weaknesses:**

**Strengths**:

The paper is well-written and easy to follow. The analyses are comprehensive and thorough.

**Weaknesses**:
1. Some expression and figures need to be revised (see below).
2. The proposed method that jointly pre-training with CL and MIM is somewhat not practical due to large training cost, though it is not the key contributions of this paper.

**Questions**:
1. In Figure1, the authors claim that *CL outperforms MIM in small model regimes*, which is inconsistent with the analyses in [a] with MAE. Some explanation should be presented.
2. Since most of the analyses are based on ViT-B, are the observations generalized to larger ViTs (*e.g.*, ViT-L or ViT-H)?
3. In Figure 3, the behaviors of MoCo are rather odd, which are also inconsistent with the observations in [b]. Are the differences from the different settings or the variance of the pre-trained models? My main concern is the reproducibility about the results. Maybe the code and weights to reproduce the results are helpful.
4. The claim in L86 that *the representations of CL contain shape information, so it can help ...* is not well-supported by the analyses on the attention distance in this paragraph.
5. I'm not sure whether L118-127 should present in Sec.2 rather than Sec.3, which focuses on the analyses of layer representations.
6. In Figure 10, it is not clear to me the meanings of "1e-1" and "1e-4" on top of the figure. And it seems that the changes for CL are larger than MIM, which appears to contradict the observations in Figure5(b) with CKA similarities.
7. The linear evaluation in L247-267 only reveals the linear separability of the representations, which could not support the claim that "Later layers of CL and early layers of MIM are important" well. Is the claim generalized well under the fine-tuning setting?

[a] Wang, S., Gao, J., Li, Z., Sun, J., & Hu, W. (2022). A Closer Look at Self-supervised Lightweight Vision Transformers. ArXiv, abs/2205.14443.

[b] Xie, Z., Geng, Z., Hu, J., Zhang, Z., Hu, H., & Cao, Y. (2022). Revealing the Dark Secrets of Masked Image Modeling. ArXiv, abs/2205.13543.

**Summary Of The Paper:**

The paper presents comprehensive analyses of self-supervised vision transformers, which provide some new insights about the differences between CL and MIM. For instance, ViTs pre-trained with CL focus on global patterns compared with MIM, collapse into homogeneity while MIM shows more diversity, and reduces the high-frequency signals of the representations but MIM amplifies them. These analyses are helpful for understanding the self-supervised ViTs and show instructions for designing new SSL methods.

**Summary Of The Review:**

The paper focuses on the self-supervised pre-trained ViTs, and presents several interesting findings, which may help to understand the different behaviors of pre-training with CL and MIM. It may guide future SSL method design and advance research in this field.

---

> ### Author Response · Authors · 2022-11-19
> **Author Response to Reviewer 54TF**
>
>
> Thank you for your thorough review and detailed comments. Below, we have addressed your concerns below.
>
> ---
>
> **Ⅰ-1. The authors claim in Figure 1-(b) is inconsistent with the analyses in MAE-lite [a].**
>
> For fair comparison, our experimental settings follow those of the original papers but with 100 epochs, and are not tailored for individual methods. MAE-lite [a] uses an increased number of backbone heads and tailored pre-training and fine-tuning settings, which seems to make a difference in performance. In addition, since MAE minimizes the decoder role of the backbone encoder (please refer to L242-246, L250-262), it may be effective even for small models with limited capacity. Thank you for suggesting this related work, and we will refer to [a] to indicate the limitation of our experiment.
>
> ---
>
> **Ⅰ-2. Are the observations generalized to larger ViTs (e.g., ViT-L or ViT-H)?**
>
> Due to the time and resource constraint, we have not been able to train ViT-{L, H} with the original settings during this discussion period ($10^3$ GPU hours for ViT-L), In addition, no large-sized pre-trained model is available from the official MoCo and SimMIM repository.
>
> Instead, we show that the observations are generalized to various sizes of ViT by investigating ViT-{Ti, S, B} as shown in Figure B.2. The results demonstrates that they have consistent self-attention behaviors. Based on this additional result, we conjecture that the observations are generalized to larger models.
>
> Moreover, we added to Figure 5 the representational similarity results of ViT-L pre-trained with MoCo using 25 epochs. The result indicates that increasing the model size does not solve or even exacerbates the lack of representational diversity problem in CL.
>
> ---
>
> **Ⅰ-3. The behaviors of MoCo in Figure 3 are inconsistent with the observations in [b].**
>
> We corrected Figure 3 since we found a mistake in MoCo data of the figure. The updated result is consistent with the observations in [b]. Sorry for the confusion.
>
> For reproducibility, we use [c]'s widely used open source implementation of self-attention distance. Our code will be released as open source along with the camera-ready manuscript.
>
> ---
>
> **Ⅰ-4. The claim in L88-90, "the representations of CL contain shape information…", is not well-supported by the attention distance.**
>
> We agree with the result of attention distance does not sufficiently support the claim – "the representations of CL contain shape information". We improve the claim in L88-90 based on the following discussions: The qualitative result (Figure 2) also shows that self attentions of CL capture the shape of objects. Fourier analysis (Figure 9) also suggests that CL exploits low-frequency information, corresponding to the shape information.
>
> ---
>
> **Ⅰ-5. I'm not sure whether L119-129 should present in Sec.2 rather than Sec.3.**
>
> Thank you for the suggestion. L119-129 presents the representational similarity experiement to explain the behaviors of CL and MIM. However, we agree that this can be confusing, and the the camera-ready manuscript will be refined taking this into account.
>
> ---
>
> **Ⅰ-6. The changes for CL in Figure 10 are larger than MIM.**
>
> Inspired by your feedback, we measure the relative displacement of representation, the ratio of the displacement to the norm of representation, and update the MoCo data. As shown in Figure F.1, the result shows that MoCo and SimMIM do not differ significantly in terms of the relative displacement. Therefore, we replace the displacement experiment with the analyzes of MAE in Figure 10. As a result, to improve the consistency of Section 4 which deals with architecture, we discuss that "the early layers of MIM and the later layers of CL are important" and that "MAE exploits such properties by using explicit decoders".
>
> ---
>
> **Ⅰ-7. Is the claim, "later layers of CL and early layers of MIM are important", generalized well under the fine-tuning?**
>
> The additional experiment shows that the claim, "later layers of CL and early layers of MIM are important", is maintained even in fine-tuning phases. To demonstrate this, we measure the drop in accuracy by fine-tuning pre-trained models initialized a few blocks from the beginning/end of the models. As we expected, the result in Figure D.1 shows that initializing the early blocks of MIM and the later blocks of CL significantly impairs the fine-tuning accuracy.
>
> ·
>
> [a] Wang, Shaoru, et al. "A Closer Look at Self-supervised Lightweight Vision Transformers." arXiv preprint arXiv:2205.14443 (2022). \
> [b] Xie, Zhenda, et al. "Revealing the Dark Secrets of Masked Image Modeling." arXiv preprint arXiv:2205.13543 (2022). \
> [c] Aritra Roy Gosthipaty and Sayak Paul, "Probing ViTs", https://github.com/sayakpaul/probing-vits (2022).

---

> > ### Comment · Reviewer_54TF · 2022-12-08
> > **Thanks for the response.**
> >
> > It seems that there have been some mistakes in the initial submission, which has also been acknowledged by the authors. All the reviewers also provide many suggestions to improve the initial submission. To include all the useful improvements and correct all the possible mistakes may require a thorough revision of the initial submission. Due to the time constraint of the reviewer-author discussion period, it may be hard for the authors to make the revision as good as it could be. Moreover, the new revision may also need a second round of review for publication. Regarding these concerns, I am inclined to degrade my initial rating to "marginally above the acceptance threshold".

---

### Public Comment · ~Yuan_Liu5 · 2022-11-10
**Problem with implementing Figure.8**

As suggested in Line#190, we take the [toolbox](https://colab.research.google.com/github/xxxnell/how-do-vits-work/blob/transformer/fourier_analysis.ipynb) to draw the log amplitude of MoCo-v3 for different depths, but we observe a quite different [trend](https://user-images.githubusercontent.com/30762564/201083840-493fcf3f-4bf2-4ff9-9c2c-314fb8b59a87.jpg). Could you please give me some advice about how to re-implementing Figure.8. Thanks in advance!

PS: the ckpt of MoCo-v3 is from [link](https://dl.fbaipublicfiles.com/moco-v3/vit-b-300ep/vit-b-300ep.pth.tar).

---

> ### Author Response · Authors · 2022-11-19
> **RE: Problem with implementing Figure.8**
>
> Hi Yuan,
>
> Thank you for the comment. We updated Figure 9 since we found a mistake in MoCo data in the figure. Sorry for the confusion. The updated result shows that the first and a few later layers of MoCo reduce high-frequency by capturing global patterns as expected. Although the high-frequency component of MoCo increases in the intermediate layers, it is generally lower than that of SimMIM. Therefore, this implies that MoCo primarily exploits low-frequency, compared with SimMIM.
>
> Best regards,
> Authors

---

### Public Comment · ~Haoqing_Wang1 · 2022-11-17
**How to calculate normalized mutual information?**

Wonderful job! I want to know how to calculate normalized mutual information. I calculate it following the paper, but I find the NMI of MoCo is not as small as shown in the paper. In fact, MoCo has the same NMI with SimMIM at lower layers but very small NMI at the top layers. So  what is the complete equation for calculate normalized mutual information? Thanks very much!

---

> ### Author Response · Authors · 2022-11-19
> **RE: How to calculate normalized mutual information?**
>
>
> Hi Haoqing,
>
> Thank you for the kind words. We revised the mutual information result. In this revision, we modified the implementation of the equation for calculating the normalized mutual information to be simpler and intuitive, and also correct MoCo data - sorry for the confusion. The updated result shows that the mutual information of MoCo is significantly smaller than that of SimMIM in the later layers and comparable in the first few layers. The code will be released as open source.
>
> Best regards,
> Authors

---

### Author Response · Authors · 2022-11-19
**Rebuttal Revision**

We appreciate the constructive feedback from all reviewers. Based on the feedback, the manuscript was revised. Some noteworthy changes include:

* We found a mistake in the MoCo checkpoint used in the following experiment: Figure 3, 4, 5, 7, 9, and 10. Most conclusions are maintained, but we updated the MoCo result in these experiment.
* We updated Figure 2 for better visualization. We use cosine similarity in Figure 5 for simplicity.
* To improve the discussion of scalability, we added the representational similarity results of ViT-{S, L} to Figure 5. We also added the self-attention behaviours of ViT-{Ti, S, B} in Figure B.2.
* To improve the discussion of the role of each component, we added linear probing result of MAE to Figure 11 and the analysis of MAE, Figure 10, to the main text.
* We added experimental results to show the claim that later layers of CL and early layers of MIM are important in find-tuning phases in Figure E.1.

---

### Decision · Program_Chairs · 2023-01-20

**Decision:**

Accept: poster

**Justification For Why Not Higher Score:**

As an empirical analysis paper, not a method one, the bar in terms of the quality of the conducted experiments is extremely high. Some of the points raised by the reviewers show that the experimental protocol could further be improved.

**Justification For Why Not Lower Score:**

The topic discussed in the paper is really interesting and will be very useful to the community. The experiments presented in the paper, despite the concerns raised in the reviews, remain strong and are sufficient to support the paper's claims.

**Metareview: Summary, Strengths And Weaknesses:**

This paper compares two broad families of SSL pre-training: Masked-Image Modelling versus Contrastive Learning. The paper proposes quantitative and qualitative evaluations shedding light on what these two learning paradigms allow. The tackled topic is timely and answers questions in which the research community focusing on this topic is currently interested.
The results are extensive, and the evidence presented is clear and reasonable. However, the reviewers pointed out some technical issues in the experimental design but tentatively answered them in the rebuttal. As the reviewers have suggested many improvements, the paper should be improved by considering them to get the paper in better shape further. I am inclined to accept this paper because of its timely message, despite some of the missing results that reviewers suggested. If the paper had to be accepted, I would like the authors to take the job of completing the draft with suggested experiments seriously and update the paper accordingly.

**Note From Pc:**

if the above contains the word "oral" or "spotlight" please see: "oral" presentation means -> notable-top-5% and "spotlight" means -> notable-top-25%. As stated in our emails, we are disassociating presentation type from AC recommendations

**Summary Of Ac-Reviewer Meeting:**

This paper was a clear borderline, with 3 6 6 8. Reviewer n5hD [3] did not join the meeting. During the meeting, we have discussed some of the flaws that reviewers have flagged and that were partially addressed by the rebuttal. The flaws and missing experiments were not judged as blockers for publication, as they did not invalidate the claims of this work. Given the timely message, we agreed on tentatively accepting this paper, as a poster, and hoping that the authors will take the reviewer's feedback into account to strengthen the draft.